# Nucleotide Sequence Variation in Long-Term Tissue Cultures of Chinese Ginseng (*Panax ginseng* C. A. Mey.)

**DOI:** 10.3390/plants11010079

**Published:** 2021-12-27

**Authors:** Sitong Liu, Xinfeng Wang, Ning Ding, Yutong Liu, Ning Li, Yiqiao Ma, Jing Zhao, Zhenhui Wang, Xiaomeng Li, Xueqi Fu, Linfeng Li

**Affiliations:** 1School of Life Sciences, Jilin University, Changchun 130012, China; liust17@mails.jlu.edu.cn; 2Ministry of Education Key Laboratory for Biodiversity Science and Ecological Engineering, School of Life Sciences, Fudan University, Shanghai 200438, China; wangxinfeng@fudan.edu.cn (X.W.); 14307110009@fudan.edu.cn (N.D.); 3Key Laboratory of Molecular Epigenetics of Ministry of Education (MOE), Northeast Normal University, Changchun 130024, China; liuyt718@nenu.edu.cn (Y.L.); lin690@nenu.edu.cn (N.L.); zhaoj878@nenu.edu.cn (J.Z.); lixm441@nenu.edu.cn (X.L.); 4Jilin Academy of Vegetable and Flower Sciences, Changchun 130033, China; xiaoquezai@163.com; 5Department of Agronomy, Jilin Agricultural University, Changchun 130118, China; wzhjlau@163.com

**Keywords:** cell totipotency, chromosome stability, nucleotide sequence variation, whole genome resequencing, tissue culture, *Panax ginseng*

## Abstract

Plants have the salient biological property of totipotency, i.e., the capacity to regenerate a whole plant from virtually any kind of fully differentiated somatic cells after a process of dedifferentiation. This property has been well-documented by successful plant regeneration from tissue cultures of diverse plant species. However, the accumulation of somaclonal variation, especially karyotype alteration, during the tissue culture process compromises cell totipotency. In this respect, Chinese ginseng (*Panax ginseng* C. A. Mey.) is an exception in that it shows little decline in cell totipotency accompanied by remarkable chromosomal stability even after prolonged tissue cultures. However, it remains unclear whether chromosomal level stability necessarily couples with molecular genetic stability at the nucleotide sequence level, given that the two types of stabilities are generated by largely distinct mechanisms. Here, we addressed this issue by genome-wide comparisons at the single-base resolution of long-term tissue culture-regenerated *P. ginseng* plants. We identified abundant single nucleotide polymorphisms (SNPs) that have accumulated in cultured ginseng callus and are retained in the process of plant regeneration. These SNPs did not occur at random but showed differences among chromosomes and biased regional aggregation along a given chromosome. In addition, our results demonstrate that, compared with the overall genes, genes related to processes of cell totipotency and chromosomal stability possess lower mutation rates at both coding and flanking regions. In addition, collectively, the mutated genes exhibited higher expression levels than non-mutated genes and are significantly enriched in fundamental biological processes, including cellular component organization, development, and reproduction. These attributes suggest that the precipitated molecular level genetic variations during the process of regeneration in *P. ginseng* are likely under selection to fortify normal development. As such, they likely did not undermine chromosomal stability and totipotency of the long-term ginseng cultures.

## 1. Introduction

Plants are sessile by nature and have evolved sophisticated developmental and phenotypic plasticity to adapt to new and changing environments [1]. Perhaps one of the most remarkable aspects of this adaptation is totipotency, referring to the ability of fully differentiated plant somatic cells to regenerate to an intact plant after a process of dedifferentiation [2,3,4,5]. Over the past decades, cell totipotency of cultured plant cells has been widely documented in diverse plant species [6,7,8]. Nevertheless, cell totipotency of cultured plant cells often declines rapidly with time in culture and often losses the trait completely within 1–2 years [9]. Mechanisms underlying this phenomenon are not well understood, but one suspected culprit is the disruption of normal cellular controls during culture, which leads to the occurrence and accumulation of genome instability, especially chromosomal instability (CIN) [10,11].

CIN includes two major aspects: numerical and structural. Numerical CIN refers to loss and/or gain of whole chromosomes or chromosomal segments, while structural CIN refers to translocation, inversion, and microscopically visible segmental gain and/or loss [12]. Conceivably CIN, being saltational changes in karyotype, may impair any biological properties especially complex ones (such as cell totipotency) that entail intricate regulation by many genes at the pathway and network levels [11]. It is, therefore, not surprising that the occurrence and accumulation of CINs would undermine totipotency or abrogate it altogether. By contrast, most molecular level mutations, such as single nucleotide polymorphisms (SNPs), are neutral, given that only a small percentage of the nucleus DNA of any higher eukaryotic organism is protein-coding genes, and even in which there is the mechanism of codon degeneracy or substantial differences in functional importance along with an open reading frame. Moreover, the two types of genome stabilities, chromosomal and molecular, are known to be controlled by distinct mechanisms [13,14]. Together, it is reasonable to assume that CIN is not necessarily coupled with molecular level genetic instability, and the later, if impacts totipotency at all, should do so at a much lower probability.

Chinese ginseng (*Panax ginseng* C.A. Meyer) is a perennial herb species within the genus *Panax* L. of family Araliaceae [15,16]. Molecular phylogeny and evolutionary history inferences have revealed a recent whole genome duplication (WGD) that occurred some 2–3 million years ago, which led to the formation of an ancestral tetraploid (2n = 4*x* = 48) followed by the splitting into three sister species, namely *P. ginseng*, *P. japonicus* and *P. quinquefolius* [17,18]. In East Asia, Chinese ginseng is widely used as a traditional herbal medicine to allegedly restore stamina and enhance the capacity to cope with fatigue and physical stress for >2000 years [19,20]. Recent studies of Chinese ginseng have also illustrated its pharmacological effects on treatment of some cancers, diabetes, cardiovascular disease, and central nervous dysfunction [21,22].

Apart from its medicinal and medical importance, *P. ginseng* also has several salient biological properties. One of these that we recently discovered [23] is the maintenance of totipotency over prolonged tissue cultures, which is accompanied by high chromosomal stability—both features are unknown in any other plant. Specifically, we found that Chinese ginseng callus subcultured for 12 consecutive years remained chromosomally stable and showed little decline in cell totipotency [23]. Further transcriptomic analyses suggested that sustained high expression of genes related to tissue totipotency and chromosomal stability might be associated with the manifestation of these two traits [23]. Nevertheless, several previous studies have shown that ginseng tissue cultures manifest mutations at the nucleotide sequence level. For example, Kiselev et al. (2013) analyzed four functionally important genes in two- and 20-y-old ginseng cultures and found that all four genes showed nucleotide variation that accumulates with culture duration [24,25]. These studies, albeit based on only a few pre-selected genes, suggest that in an aspect of molecular level genetic variation under tissue culture, ginseng is perhaps not much different from other plants. Nonetheless, to have a definitive answer to this matter, genome-wide analyses are needed, which are currently lacking.

Here, we addressed this issue based on findings reported in our previous study [23]. Specifically, we performed whole-genome resequencing and conducted genome-wide, single base-resolution analyses for molecular level genetic variations in plants regenerated from long-term cultured calli that were still highly totipotent and showed little CIN [23]. We characterized the SNPs with respect to their relevance to genomic features, chromosomal distribution, gene category, and expression. We conclude that molecular level genetic variation occurred in prolonged tissue cultures of *P. ginseng*, and which were retained in regenerated plants. However, these molecular level mutations may not have affected chromosomal stability and totipotency.

## 2. Methods and Materials

### 2.1. Plant Materials and Callus Culture

The original ginseng explant was taken from a single bud of *P. ginseng* cv. Damaya in 2004 [23]. The primary callus was induced with a standard Murashige–Skoog (MS) solid medium containing 2 mg/L 2,4-dichlorophenoxyacetic acid (2,4-D) at 26 °C under dark conditions for 1 month. Then, the newly formed embryogenic calli were selected and subcultured with MS medium under the same conditions (at 30-day intervals) for 12–13 years [23]. The embryogenic calli remained totipotent, and in 2016 and 2017, they were transferred to a regeneration medium containing MS basal + 0.3% hydrolyzed casein acid + 0.1% proline + 1 mg/L 2,4-D and culturing at 26 °C under 14/10 light/dark conditions [26]. The regenerated ginseng plantlets were taken for subsequent experiments.

### 2.2. DNA Extraction and Whole Genome Resequencing

The first batch of samples included 20 randomly selected ginseng plantlets regenerated from 12-y-old ginseng calli, and which were arbitrarily divided into two pools. The second patch of the sample included seven regenerated plantlets from 13-y-old ginseng calli. DNA extraction of the three pooled samples were conducted using a modified CTAB method [27]. The high-quality genomic DNA of each sample was utilized for subsequent whole-genome resequencing. Short pair-end (150 bp) DNA libraries of the three samples were constructed with Illumina Trueseq DNA PCR-Free kit at Novogene (Tianjin, China) and sequenced with the Illumina Hi-seq 4000 platform (Illumina, CA, US). The data were deposited in the SRA database of GenBank (http://www.ncbi.nlm.nih.gov/sra, accessed on 20 November 2021) with the BioProject accession number PRJNA782439. Only clean reads (base quality > 30) were retained for subsequent data analyses.

### 2.3. Identification of De Novo Genetic Variants and Their Chromosomal Distribution

To identify genetic variants accumulated in these tissue culture-regenerated plant samples, clean short reads of the three pooled samples were mapped to the *P. ginseng* reference genome using Burrows–Wheeler Aligner (BWA) [28], with default parameters. Then, the program Samtools [29], Genome Analysis Toolkit (GATK) [30] and Picard (http://broadinstitute.github.io/picard/, accessed on 19 November 2021) were used to sort, index, and realign the mapped short reads, and remove duplicates. To identify SNPs with high confidence, the realigned short reads were further filtered using the program Bcftools [29], with the parameters “-a DP,ADF,ADR -Q 30 -q 30 -Ou”. Low-quality SNPs with base quality (Q) or mapping quality (q) smaller than 30 were excluded from the data analyses. In addition, we performed additional quality control of the SNPs according to the following procedures: (1) each allele of the heterozygous (Ht) SNPs possessing > five mapped reads; (2) the homozygous (Ho) SNPs with read depth ≥5 were retained; (3) SNPs with multi-allelic (≥3) variants, insertions and deletions were excluded. Only those SNPs that passed the above quality controls were used for further analyses. Based on these identified Ht SNPs, we further defined the de novo variants as those that were non-shared among the three samples (y_2016_1, y_2016_2 and y_2017), which should be variations that occurred during tissue culture and independently retained in the regenerated plants.

### 2.4. Functional Analyses of Mutated Genes

The program SnpEff [31] was used to annotate the identified SNPs from the three pooled samples independently. Then, we calculated the number of non-synonymous (Sn) and synonymous (Ss) mutations for each gene. Based on this annotated information, all genes were divided into two categories, namely mutated gene (containing SNPs) and non-mutated gene (without SNPs). The mutated gene was further divided into two types, Sn gene (containing Sn mutations) and the other type (no Sn mutations). Functions of genes in different classes were analyzed and inferred using WEGO 2.0 (Web Gene Ontology Annotation Plot, http://wego.genomics.org.cn/, accessed on 21 November 2021) [32]. Gene expression data and genomic pattern of genes in different classes were retrieved from our previous study [23]. Distribution patterns of the identified SNPs were visualized using R package “ggplots” and “RColorBrewer”. Correlation between gene mutation and expression level was analyzed in R package “ggplots”.

## 3. Results

### 3.1. Identification of Molecular Level Genetic Mutations in Ginseng Plantlets Regenerated from Long-Term Callus

Whole-genome resequencing of the three pooled ginseng callus-regenerated plantlet samples generated a total of 1,638,664,104 clean pair-end Illumina reads (reads length = 150 bp), with each sample possessing 36–58x genome coverage (Appendix A). Based on this deep sequencing data, we identified a total of 1,480,353 Ht SNPs from the three samples, each containing from 336,972 to 1,137,450. Among these nucleotide variants, 343,330 were common to all three samples (Table 1). Given that (1) the ginseng plants are known as genetically heterogeneous among individuals even within the same cultivar (e.g., Damaya), and (2) we no longer had the exact plant individual (as the control) from which the explant was taken to initiate the callus cultures, we assigned SNPs shared by all three samples as standing genetic differences between our explant individual and the ginseng reference genome. As such, only the 1,137,023 Ht SNPs that are non-shared among the three samples were defined as de novo variants that occurred during the callus culture process and were retained in the regenerated plantlets (Table 1). Although, to an extent, this conservative treatment may lead to an underestimate of the culture-induced variations, it confidently ruled out false positives. Albeit potential underestimated, our data clearly show that somaclonal variation in the form of SNPs definitively occurred in the long-term cultured ginseng callus.

Next, we interrogated how these de novo genetic variants were associated with genomic features in the ginseng genome. At the whole genome level, the y_2017 sample (0.34 Ht/Kb) contained higher nucleotide diversity compared to both of the two y_2016 samples (0.18 and 0.10 Ht/Kb). At the genic region, all three samples showed slightly lower nucleotide diversity (0.09–0.26 Ht/Kb) relative to the whole genome level. Notably, the cell totipotency- (0.06–0.16 Ht/Kb) and chromosome stability-related genes possessed obviously lower nucleotide diversity at genic regions compared to the genome-wide overall genic regions. We also calculated the number of Ss and Sn mutations in the three samples (Appendix A). A general pattern is that all three samples contained more Sn than Ss mutations for the overall and cell totipotency/chromosome stability-related genes. Among the three samples, the y_2017 sample (Ss = 8666 and Sn = 16,806) harbored higher numbers of Ss and Sn mutations compared to the two y_2016 callus samples (Ss = 3851–6699 and Sn = 7271–12,691) for the overall genes. Likewise, we also identified smaller numbers of Ss and Sn mutations in the cell totipotency-related genes (Ss = 3.95–15.30 × 10^−6^ and Sn = 9.37–25.20 × 10^−6^) and chromosome stability-related genes (Ss = 8.62–24.80 × 10^−6^ and Sn = 13.90–45.60 × 10^−6^) than in those of the overall genes (Ss = 12.10–27.10 × 10^−6^ and Sn = 22.80–52.60 × 10^−6^).

### 3.2. Spectrum and Trend of the Molecular Level Genetic Mutations

The above analyses revealed distinct levels of nucleotide diversity among the three ginseng callus-regenerated plantlet samples and differences in different genomic regions. We then calculated the spectrum of the genetic variants along the 24 ginseng chromosomes (Figure 1). Results showed that these genetic variants were not randomly distributed along a given ginseng chromosome, and the amount of mutations differed among the chromosomes. For example, each of the 24 chromosomes possessed several high-mutation genomic regions (referred to as mutation clusters) across the three samples. In addition, some chromosomes (i.e., Chr06) harbored more mutation clusters than the others (i.e., Chr24). In particular, all the homeologous chromosome pairs of the two subgenomes differed in the number and distribution pattern of mutation clusters. Broadly, all three samples showed similar distribution patterns of mutation clusters along the 24 chromosomes.

We next estimated the spectrum of Ss and Sn mutations separately along each chromosome of the three samples (Appendix A). Similar to the mutation spectrum at the overall level, both Ss and Sn variants also exhibited distinct variation patterns across the 24 chromosomes. For example, both the number and distribution of mutation clusters differed between the Ss and Sn mutations within and between the 24 chromosomes. In addition, our results also revealed that the mutation clusters trended towards the telemetric regions (Appendix A). However, an opposite distribution pattern was observed for the overall genetic variants (see Figure 1). It is notable that while the three samples showed a highly similar overall distribution pattern of these genetic variants, they possessed some sample-specific mutation clusters of all the three types of genetic variants (overall, Ss and Sn), especially those on the centromeric region of Chr06.

### 3.3. Associations between Genetic Variation and Gene Expression

The above analyses revealed clustered distribution of the genetic variants. We next examined whether the mutated genes would show different transcription levels compared to the non-mutated (wild-type) genes. A general pattern observed at the overall level was that genes localized at distal (subtelomeric) regions showed higher expression levels compared to those of at the interstitial (pericentromeric) regions, although each of the 24 chromosomes showed different overall transcription levels (Figure 2a). Between the two subgenomes, the 12 homeologous chromosome pairs possessed highly similar expression pattern at the distal regions. For example, genes localized at the two distal genomic regions of the homologous chromosome pair 1 and 9 showed distinct transcription levels. Further analysis of the relationship between gene transcription and genetic variants, we found that the mutated genes possessed higher transcription levels than those of non-mutated (i.e., wild-type) genes in all three samples (Figure 2b). Furthermore, mutated genes containing non-synonymous mutations showed relatively higher expression levels compared to genes with synonymous mutations, although both exhibited higher transcription levels relative to the non-mutated genes.

### 3.4. Functional Enrichment of the Mutated Genes

All genetic variants identified in the three regenerated plantlet samples should have occurred during the callus subculture process. We then asked the question: were these mutated genes functionally associated with chromosome stability and cell totipotency? Functional enrichment analyses revealed that the cell totipotency- and chromosome stability-related genes showed distinct enrichment patterns compared to the total genes (Figure 3). At the cellular component level, for example, compared to the cell totipotency-related genes and total genes, the chromosome stability-related genes showed relatively higher enrichment in the gene ontology (GO) terms related to intracellular cell, organelle part, and supramolecular complex. A similar pattern was also observed at the molecular function level where the chromosome stability-related genes exhibited higher enrichment in protein activity, molecular and chemical compound binding. At the biological process, however, the cell totipotency-related genes exhibited higher percentages of enrichment in these GO terms involved in diverse metabolic processes, plant development, and stress responses.

Based on the above functional analyses at the overall level, we further examined whether a similar enrichment pattern existed in the mutated genes. Our results revealed that mutated genes of the three samples showed different GO term enrichments compared to the overall genes, as detailed above (Appendix A). In the cellular component category, the mutated genes of sample y_2016_1 exhibited a similar enrichment pattern to the overall genes, with the majority of the enriched GO terms related to intracellular cell and organelle part. However, the other two samples (y_2016_2 and y_2017) were enriched to the other cellular components, such as the membrane part and supramolecular complex. In contrast, the three samples exhibited moderate differences in the functional enrichments in the molecular function and biological process categories. For example, compared to the non-mutated genes, mutated genes of the three samples showed functional enrichment in biological processes related to metabolic process, development, reproduction, and stress responses. In the molecular function category, the mutated genes were mainly enriched in functions related to protein activity and molecular binding.

It is notable that the mutated and non-mutated genes showed different functional enrichments. In the cellular component category, the chromosome stability-related genes exhibited a higher degree of enrichment in components related to intracellular, organelle and supramolecular complex (Appendix A). However, the mutated genes containing Ss and Sn mutations exhibited lower degrees of functional enrichment than the non-mutated genes. Yet, the mutated genes possessed similar enriched GO terms to the cell totipotency- and chromosome stability-related genes in the biological process and molecular function categories. The synonymous and non-synonymous genes exhibited higher degrees of enrichment than the non-mutated genes, particularly those related to cellular component organization, plant development, and reproduction. In addition, we also examined the mutation types for the cell totipotency- and chromosome stability-related genes. A general pattern was that majority of these important functional genes possessed multiple copies in the ginseng genome (Appendix A). For example, 108 of the 388 cell totipotency-related genes were functionally related to auxin efflux carrier and response in the ginseng genome (Appendix A). Likewise, we also identified 88 casein kinase1 (CK1) and 102 CGMC (CGMC includes CDA, MAPK, GSK3, and CLKC kinases) in the 533 chromosome stability-related genes (Appendix A). It is notable that, among the 388 cell totipotency-related genes, some functionally important genes possessed both the Sn and Ss mutations, including those involved in auxin response and DNA methylation. Similarly, we also identified Sn and Ss mutations in some of the 533 chromosome stability-related genes, such as dnaK, minichromosome maintenance subunit 2/3(MCM2/3), and CK1.

## 4. Discussion

Plants possess a generally stronger capacity to regenerate new tissues and organs upon loss or injury of body parts under natural conditions than animals [33,34,35]. Plant in vitro tissue culture experiments have uncovered the even more remarkable property of plants, i.e., totipotency, referring to the phenomenon that fully differentiated plant somatic cells are capable of re-differentiating into intact plants after a de-differentiation phase [36,37,38,39]. However, during tissue culture, diverse genetic and epigenetic instabilities (collectively termed somaclonal variation [10]) occur, which on the one hand may comprise the elite phenotypes of the donor plants while on the other hand can be exploited for beneficial variants [9,13]. While tissue culture-induced epigenetic variation mainly involves changes in DNA methylation, that is, loss/gain of methylated cytosine bases [40], genetic variation is more complex and can be broadly categorized into chromosomal- and sequence-level mutations [41]. It is generally believed that somaclonal variation is a major cause of the rapid decline and/or loss of totipotency in plant tissue cultures [42].

We reported recently that, different from other plant species, a cultured ginseng somaclonal callus line showed little decline of totipotency even after >12 years in subculture [23]. Of note, our study showed that this ginseng somaclonal callus line maintains high chromosomal stability in both number and structure. As far as we know, these two properties of *P. ginseng* have not been observed in any animals (with respect to somatic chromosomal stability) and other plants. However, previous studies have shown that mutations at the nucleotide sequence level occurred in ginseng tissue cultures, with mutation rate increases in a time-dependent fashion [24,25]. It thus seems that ginseng was not particularly different from other plants in this aspect, which appears incongruent with our results [23]. However, chromosomal and nucleotide sequence instabilities are known to be controlled by distinct pathways [13,14,43]. Therefore, it is possible that sequence-level mutations also occurred in the callus somaclonal line we studied [23], which did not undermine totipotency. If this were the case, it also would be interesting to learn what features these mutations have.

Here, we investigated this matter based on genome-wide resequencing and analyses of ginseng plantlets regenerated from the same somatic callus line that showed remarkable chromosomal stability [23]. We uncovered numerous genetic variants at the nucleotide sequence level, although we have adopted a highly conserved treatment of the data, i.e., excluding the variants that are shared by all three samples, each containing a pool of regenerated plantlets (see Results). We found several features of these mutations, including (i) emergence of mutation clusters and their biased distribution towards distal chromosome regions; (ii) higher non-synonymous than synonymous mutation rates; (iii) lower mutation rates in cell totipotency- and chromosomal stability-related genes than overall genes; (iv) higher expression levels in mutated genes than non-mutated genes; and (v) enrichment of mutated genes to key basic biological processes including cellular component organization, development, and reproduction.

Notably, because we analyzed regenerated plantlets rather than callus per se, the mutations we detected should be a small fraction of those that occurred in the callus, and, which were retained in the regeneration process. Thus, there is no doubt that mutations at the nucleotide sequence level did occur in cultured ginseng cells, consistent with prior studies [24,25]. The features we summarized above may suggest that mutations in our long-term ginseng cultures did not occur at random. Moreover, they were likely under both purifying and positive selections. First, because distal chromosome regions are known as recombination-active, if somatic recombination events [44] had occurred in the long-term culture process, then it is possible that deleterious mutations can be purged out. Second, both higher non-synonymous than synonymous mutation rates and higher expression levels of mutated than non-mutated genes suggest the possibility of selection. Third, lower than average mutation rates in genes related to cell totipotency and chromosomal stability suggest either non-random occurrence of mutation or more efficient repairing for these critical genes. Finally, specific enrichment of mutated genes to fundamental biological processes may suggest their potential roles to compensatory for maintaining critical cellular homeostasis and to fortify normal development. Together, these features of the mutations we uncovered in the long-term ginseng cultures strongly suggest that they did not undermine chromosomal stability nor compromise cell totipotency. For example, several genes related to auxin response, i.e., auxin response factor 19 and auxin efflux carrier component, possess numerous non-synonymous mutations. It has been documented that genes involved in auxin and cytokinin biosynthesis play important roles in the cell totipotency of cultured plant cells [1,34]. This suggests that these genetic mutations are potentially associated with the maintaining of cell totipotency in the long-term tissue culture process.

## Figures and Tables

**Figure 1 plants-11-00079-f001:**
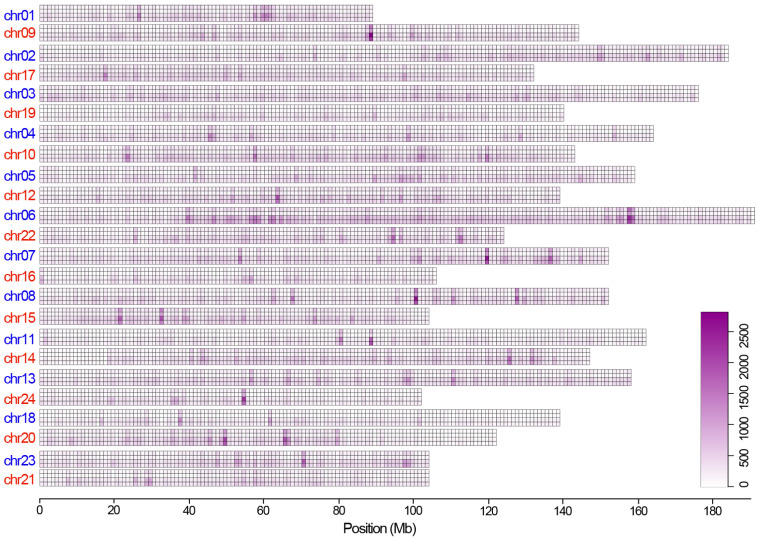
Genomic/chromosomal landscape of the genetic variants identified in the three pooled ginseng callus-regenerated plantlet samples. The four rows from top to bottom for each chromosome are genetic variants identified in samples y_2016_1, y_2016_2, y_2017, and de novo genetic variants. The numbers of the four types of Ht SNPs are the same as in Table 1. Homeologous chromosomes of the two ginseng subgenomes are labeled in blue and red, respectively.

**Figure 2 plants-11-00079-f002:**
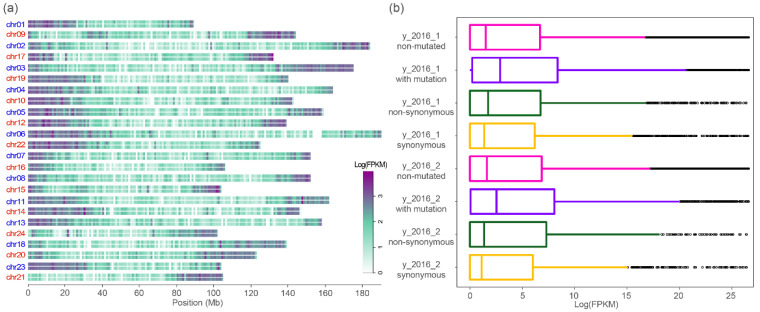
Genome-wide gene expression pattern and correlation with a genetic mutation. (**a**) The transcription level of the total genes along each of the 24 chromosomes. The two rows for each chromosome correspond to the sample y_2016_1 and y_2016_2, respectively; (**b**) comparisons between the overall gene expression levels and the types of genetic variants.

**Figure 3 plants-11-00079-f003:**
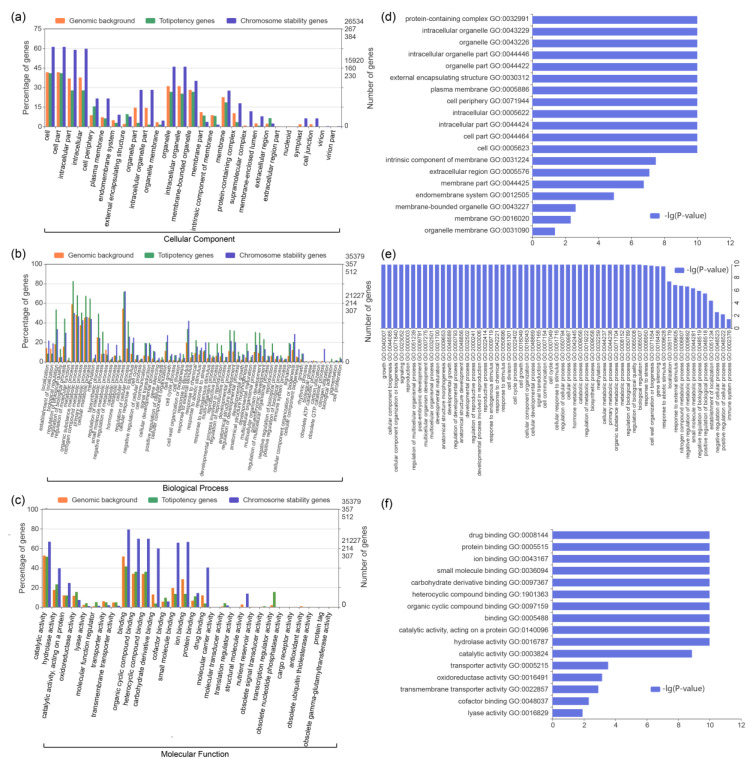
GO enrichment analysis of the total totipotency and chromosome stability-related genes. Enriched GO terms in cellular component (**a**,**d**), biological process (**b**,**e**) and molecular function of the total genes (**c**,**f**), totipotency genes and chromosome stability genes.

**Table 1 plants-11-00079-t001:** The number of heterozygous (Ht) genetic variants and nucleotide diversity in three pooled callus-regenerated plantlets of *Panax ginseng*.

Items	y_2016_1	y_2016_2	y_2017	Total Ht SNPs *	Ht SNPs Shared by All Three Samples ^#^	De Novo Ht SNPs in the Three Samples ^&^
Genome-wide heterozygotes	591,204	336,972	1,137,450	1,480,353	343,330	1,137,023
Genome size (bp)	3.36 × 10^9^	3.36 × 10^9^	3.36 × 10^9^	3.36 × 10^9^	NA	3.36 × 10^9^
Average genome-wide Ht (per Kb)	0.18	0.10	0.34	0.44	NA	0.34
Genic heterozygotes	55,308	28,718	83,996	118,949	27,347	91,602
Length of all genes (bp)	3.20 × 10^8^	3.20 × 10^8^	3.20 × 10^8^	3.20 × 10^8^	NA	3.20 × 10^8^
Genic Ht (per Kb)	0.17	0.09	0.26	0.37	NA	0.29
Totipotency-related genic heterozygotes	174	114	329	450	98	352
Length of all totipotency- related genes (bp)	2.03 × 10^6^	2.03 × 10^6^	2.03 × 10^6^	2.03 × 10^6^	NA	2.03 × 10^6^
Totipotency-related genic Ht (per Kb)	0.09	0.06	0.16	0.22	NA	0.17
Chromosomal stability- Related genic heterozygotes	563	244	875	1272	257	1015
Length of all chromosomal stability- genes (bp)	3.94 × 10^6^	3.94 × 10^6^	3.94 × 10^6^	3.94 × 10^6^	NA	3.94 × 10^6^
Chromosomal stability- related genic Ht (per Kb)	0.14	0.06	0.22	0.32	NA	0.26

* Total Ht SNPs are heterozygous SNPs in the three samples that are different from the reference genome; ^#^ these Ht SNPs are fixed in all three samples compared to the reference genome; ^&^ these Ht SNPs are different among the three samples, which are assumed to occur during the tissue culture process; NA., no information is available.

## Data Availability

All Illumina data generated in this study has been deposited to SRA under the accession number of PRJNA782439.

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
