# Peer review of "Nucleotide Sequence Variation in Long-Term Tissue Cultures of Chinese Ginseng (Panax ginseng C. A. Mey.)"

_plants, 2021, doi:10.3390/plants11010079_

Round 1

Reviewer 1 Report

You tried to detect the variations at the plant level. Plants regenerate from normal cell usually. While callus will have more variations and some of which will inhibit regeneration. In such cases how one knows about the changes that have occurred.

Author Response

Reviewer 1

Comments and Suggestions for Authors

You tried to detect the variations at the plant level. Plants regenerate from normal cell usually. While callus will have more variations and some of which will inhibit regeneration. In such cases how one knows about the changes that have occurred.

Reply: We appreciate this very pertinent comment, and to which we fully agree! Indeed, we analyzed variations that occurred in the callus but RETAINED in the regenerated plants. As stated in the Abstract, we previously found that the ginseng callus in our hands is strikingly different from all other plants studied in that it remained totipotent and chromosomally high stable even after 12 years of culture. As an extension to that prior study, the objective of this study is to investigate whether these salient features of ginseng are necessarily entail molecular level genetic stability. Our results demonstrated that it is not necessarily so. As stated in our manuscript, the variations we detected are certainly an underestimate of the variations that occurred in the callus. However, our strategy in analyzing he regenerated plant instead of the callus per se can fully address the question we asked. Again, we fully agree with this comment. If for the purpose to accurately assess the extent/spectrum of variations that could occur in the callus (rather than if variations occurred), then direct analysis of the callus itself is needed, which however would require different study strategies and much deeper sequencing given the intermingled clonality and high heterogeneity of the cells, just like the situation of human cancer cells. Per this comment, we further clarified the point in the revised manuscript (Page 3, Lines 140-142).

Reviewer 2 Report

Kindly revise the  manuscript in the light of these comments:

  1. Second page, Line 79"issue culture", correct it.
  2. At places authors have used non-mutated ( line 226)for wild-types, while at some places " none-mutated" term has been used( line 235), better to use 'non-mutated" only. Correct in whole MS.
  3. Line 245-246, "We then...totipotency", reframe the sentence, use question mark while asking a question. Same with Line 325.
  4. Line 345" somatic recomination evens"  ?
  5. Please check the manuscript for grammar and proper punctuations , which are missing at places.
  6. The study is very elaborated, nicely conducted but it has not been mentioned that is it the first of its kind? Has such study been conducted in any other plants? This will highlight your work.

Author Response

Reviewer 2

Comments and Suggestions for Authors

  1. Second page, Line 79"issue culture", correct it.

Reply: Corrected. Thanks!

  1. At places authors have used non-mutated (Line 226) for wild-types, while at some places " none-mutated" term has been used (Line 235), better to use 'non-mutated" only. Correct in whole MS.

Reply: Corrected all these typos. Thanks!

  1. Line 245-246, "We then...totipotency", reframe the sentence, use question mark while asking a question. Same with Line 325.

Reply: Made changes accordingly for lines 245-246 (new version 273-274). For line 325 (new version 364), however, we considered our phrasing reflects our intended meaning. Thanks!

  1. Line 345 " somatic recombination evens"?

Reply: It is “somatic recombination events”. Corrected. Thanks!

  1. Please check the manuscript for grammar and proper punctuations, which are missing at places.

Reply: We carefully went through our manuscript. All the typos were corrected. Thanks!

  1. The study is very elaborated, nicely conducted but it has not been mentioned that is it the first of its kind? Has such study been conducted in any other plants? This will highlight your work.

Reply: We appreciate this comment! Yes, with respect to genome-wide analysis of somaclonal variation, this is the first work in Panax ginseng. However, in other plants such as Arabidopsis, rice and maize, this type of studies have been previously. Notably, however, P. ginseng is an exception in that it shows little decline in cell totipotency accompanied with remarkable chromosomal stability even after prolonged tissue cultures. Per your comment, we emphasized this important feature of ginseng tissue calli (Page 9, Lines 336-339).

Reviewer 3 Report

# comment-1

The analysis part of material and methods is too short. Please explain in more detail how the figures were produced. For example for the distribution pattern of mutations and expression levels, box plots, etc. I suppose the authors used different R packages for some figures and it is not appropriate to use packages without citation.

# comment-2

In figure 2b,  for box plots it is better to use the word “comparison” instead of “correlation”.

# comment-3 English language is fluent and only few typos correction is required

Author Response

Reviewer 3

Comments and Suggestions for Authors

# comment-1

The analysis part of material and methods is too short. Please explain in more detail how the figures were produced. For example for the distribution pattern of mutations and expression levels, box plots, etc. I suppose the authors used different R packages for some figures and it is not appropriate to use packages without citation.

Reply: We appreciate this suggestion! We added the related content in the revised manuscript (Page 3, Lines 129-142; Page 4, Lines 158-161).

# comment-2

In figure 2b, for box plots it is better to use the word “comparison” instead of “correlation”.

Reply: We made changes accordingly. Thanks!

# comment-3

English language is fluent and only few typos correction is required.

Reply: We appreciate this positive comment! We checked the manuscript again and corrected the typos.